# Determinants of Sleep Disorders and Occupational Burnout among Nurses: A Cross-Sectional Study

**DOI:** 10.3390/ijerph19106218

**Published:** 2022-05-20

**Authors:** Agnieszka Młynarska, Magdalena Bronder, Ewelina Kolarczyk, Stanisław Manulik, Rafał Młynarski

**Affiliations:** 1Department of Gerontology and Geriatric Nursing, Faculty of Health Sciences, Medical University of Silesia, 40-635 Katowice, Poland; katedrapielegniarstwa@sum.edu.pl; 2Department of Clinical Nursing, Faculty of Health Sciences, Wroclaw Medical University, 51-618 Wroclaw, Poland; stanislaw.manulik@umed.wroc.pl; 3Department of Electrocardiology, School of Health Sciences, Medical University of Silesia, 40-635 Katowice, Poland; rmlynarski@sum.edu.pl

**Keywords:** sleep disorders, nurses, shift work, burnout

## Abstract

Background: The aim of the study was to assess the determinants of the sleep disorders that occur among nurses working in a shift system by assessing the influence of sociodemographic factors, the impact of shift work, and the occurrence of occupational burnout. Methods: The study included 300 nurses who work shifts in the Silesian Region (Poland). The research was conducted using standardized research tools: the Karolinska Sleepiness Scale (KSS), the Epworth Sleepiness Scale (ESS), the Athens Insomnia Scale (AIS), and the Maslach Burnout Inventory (MBI). Results: Among the sociodemographic factors, in the KSS analysis, sleep disorders were most common in men (CI: 0.038; *p* < 0.001), in divorced individuals (CI: 1.436; *p* = 0.045), and in individuals who were overweight (CI: 1.927; *p* = 0.026). Multiple linear regression showed that sleep disorders (*p* < 0.001) were an independent predictor of MBI among nurses who worked shifts. Conclusions: Sleep disturbances affect the burnout of nurses who work shifts.

## 1. Introduction

Sleep is one of the most important elements affecting the regeneration of the human body after all-day activity [1]. Sleep is necessary to maintain the balance between the states of the body and mind; today, this balance is significantly disturbed, as there is an increasing tendency to shorten sleep time. Sleep disorders are among the most common health disorders [2], and the problem of insomnia affects people of all ages, but it mostly affects workers who work shifts [3]. It is estimated that approximately 20–30% of shift workers experience pronounced symptoms of insomnia and excessive daytime sleepiness as a result of a sleep disturbance associated with disturbed circadian rhythms related to shift work [4]. This is caused by an excessive number of professional duties, mental effects, and coexisting somatic diseases, as well as the intensification of life, excessive haste, an increasing number of professional and family obligations, and the desire to increase one’s social status [5]. Despite the fact that sleep disorders affect the entire population regardless of age and position in society, they are observed more often during the period of later maturity and in people with a greater number of working years [6].

As reported in the literature, some professional groups are definitely more exposed to this phenomenon. Work in night and shift modes, accompanied by excessive stress, makes it very difficult to rest and sleep properly. This leads to many diseases, including hypertension, diabetes, ischemic heart disease, anxiety disorders, neurosis, and depression [7,8,9], which are the cause of numerous sick leaves, professional burnout, and conflicts with the environment and with relatives. A tired person, who is deprived of the possibility of rest at night, becomes frustrated, and their work is less efficient, which causes a greater risk of accidents and other undesirable situations [10]. Occupational burnout is a psychological syndrome of emotional exhaustion, depersonalization, and reduced sense of personal achievement in people working with others [11]

Performing work at night is contrary to the natural biological rhythm of a human being and has a negative impact on one’s performance and health [12]. Night work worsens the wellbeing of the employee and accelerates the accumulation of fatigue, which is the cause of a lower quality and efficiency of work and numerous absences of the employee [13,14]. Shift work-related sleep disorders mainly manifest themselves as insomnia or excessive sleepiness and occur as a temporary problem with the distribution of working time. Changing one’s circadian rhythm results in a significant inability to maintain sleep, which is shifted to the daytime. Resting at times other than at night results in poor sleep quality, even though an individual gets sufficient sleep during the day [15,16]. Many studies have shown that proper sleep quality and proper sleep should be the overarching goals in our daily lives [17]. Currently, mainly in developed countries, there is a growing phenomenon with respect to the number of shift workers, primarily in the medical sector [18]. Little is known about the problem of sleep disorders in relation to occupational burnout in nurses who work in shifts. There is still an insufficient understanding of the sociodemographic factors that determine sleep disorders that can also affect the burnout in shift-working nurses.

The main aim of the study was to assess the determinants of sleep disorders among nurses. The detailed objectives of this study included an assessment of the impact of sociodemographic factors on sleep disorders, the impact of shift work on sleep quality and efficiency, the wellbeing of the respondents after a sleepless night, and the occurrence of burnout in the context of sleep disorders. This knowledge is important to improve nursing practice.

## 2. Materials and Methods

### 2.1. Study Design

A cross-sectional survey method was used in the research. The study included 300 nurses who work in a shift system and who live and work in the Silesian Region in Poland. The minimum sample size was 266, which was calculated on the basis of the available nursing population with a 95% confidence interval. The data to determine the minimum number of individuals were obtained from a report by the Supreme Chamber of Nurses in Poland. The data for the present study were collected from July to November 2019 using a paper-and-pencil questionnaire. In nursing, there are many varieties of shift work organization systems, such as three-shift, two-shift, and 10 h shift regimes. In many clinics, nurses work in one shift, i.e., from Monday to Friday in equal hours. In turn, section nurses working in hospitals usually perform their work in a two-shift system. This is characterized by two 12 h shifts (day and night). Nurses start work at 7:00 a.m. or 7:00 p.m. On average, they work in one shift for 1 day, with 2 days off followed by 2 days on [4,5]. The criteria for inclusion in the study were being as a nurse and undertaking shift work. The criteria for not being included in the study were a lack of consent to participate in the study and an incomplete survey.

### 2.2. Instruments

The research was conducted using standardized research tools: the Karolinska Sleepiness Scale (KSS), the Epworth Sleepiness Scale, the Athens Insomnia Scale, and the Maslach Burnout Inventory (MBI).

The Karolinska Sleepiness Scale questionnaire enables the degree of sleepiness during the day and the likelihood of falling asleep during the day to be determined. The questionnaire is intended for individuals who struggle with excessive sleepiness during daily activities, and it is a single-item measure of sleepiness (1 = “very alert” to 9 = “very sleepy great effort to keep awake, fighting sleep”) [19,20]. For this study, a high level of sleepiness during the shift was defined as a score >7 (upper one-third of the scale range). The KSS has been widely used and has provided reasonable results in studies of shift work, driving abilities, attention and performance, and clinical settings [21].

The Epworth Sleepiness Scale (ESS) was developed in 1990 as a measure of ‘daytime sleepiness’ for adults [22]. It is used for self-assessment and consists of determining the possibility of falling asleep in eight everyday situations by the respondent [23]. The scale has three classes depending on the presence and intensity of daytime sleepiness: I—norm, for scores in the range of 0–10 points; II—moderate daytime sleepiness, for scores in the range of 11–15 points; III—increased daytime sleepiness, for scores above 16 points. The ESS is widely used throughout the world for clinical and research purposes [24].

The Athens Insomnia Scale enables an assessment of whether the person completing the questionnaire experiences insomnia. The questionnaire consists of eight questions that are scored from 0 to 3 points. A sum of points equal to or higher than 6 indicates insomnia (scale sensitivity—93%, scale specificity—85%) [25]. The high measures of consistency, reliability, and validity of the AIS make it an ideal tool in sleep research and clinical practice. The internal consistency (Cronbach’s alpha = 0.90) and the test–retest reliability (*r^2^* = 0.92) of the AIS were found to be very satisfactory [26].

The Maslach Burnout Inventory (MBI) enables the level of occupational burnout in three aspects (emotional exhaustion, depersonalization, and job dissatisfaction) to be assessed. MBI was developed by Maslach and Jackson in 1981. The MBI was validated in Polish by Pasikowski [22] and achieved the following Cronbach’s alpha coefficients for the scales: 0.85 for EE, 0.60 for DEP, and 0.76 for PA. The scores on each of these subscales are given on a 0–100 scale, where a higher score indicates a higher level of occupational burnout. Moreover, the general index of occupational burnout, which is an average of the three subscales, is also calculated. For the versions with yes/no answers, there are standards that enable it to be determined whether the burnout in the respondent is strong or not [27]. The MBI questionnaire, which has been used to measure occupational burnout among nurses worldwide, is a useful tool for determining the effectiveness of burnout reduction measures recommended in health policy planning [28,29].

Additionally, a proprietary demographic and epidemiological questionnaire was used, which included questions about the individual’s economic situation, employment, and current health condition.

### 2.3. Statistical Analyses

The obtained results were systematized and compiled in terms of quantity and quality. In the analysis of the quantitative variables, the mean, standard deviation, median, quartiles, minimum, and maximum were used. The qualitative variables were analyzed by calculating the number and percentage of the occurrences of each value. The differences in the two groups were determined using the Mann–Whitney test. The values of the variables in three or and more groups were compared using the Kruskal–Wallis test. When statistically significant effects were observed, a post hoc analysis using Dunn’s test was performed to identify the statistically significantly different groups. The relationships between the quantitative variables were analyzed on the basis of the Spearman correlation coefficient. The multivariate analysis of the independent influence of many variables on the quantitative variable was performed using the linear regression method. The results are presented in the form of the values of regression model parameters with a 95% confidence interval. The level of significance in the analysis was *p* < 0.05. Statistical calculations were performed using the R computer program, version 3.6.1.29 [30].

### 2.4. Ethical Procedure

All of the participants were informed of the detailed course of the study, its purposefulness and anonymity, and the method of filling in the questionnaire. In addition, the respondents were informed about the possibility of withdrawing from participation in the study at any of its stages without suffering any consequences. All of the respondents consented to participate in writing, and the study protocol was approved by the Independent Bioethics Committee of the Silesian Medical University in Katowice (decision no. PCN/0022/KB/15/20). The research was conducted in accordance with the recommendations of the Helsinki Declaration developed by the World Medical Association [31] and the guidelines of Good Clinical Practice [32].

## 3. Results

A total of 300 respondents took part in the survey. Most respondents (264/300—88.00%) were women, and the mean age was 39.49 ± 8.87 years. The vast majority of the respondents had a normal body weight (158/300—52.67%), had completed higher education (135/300—45.00% and 133/300—44.33%), had children (209/300—69.67%), and were in a relationship or were married (203/300—67.67%). The Kruskal–Wallis test showed that marital status was significantly related to sleep disorders according to the Athenian ASI scale (*p* = 0.018) and the Epworth ESS scale (*p* = 0.004). Only nine (54/300—25.84%) of the respondents had preschool children. The most numerous group of individuals with chronic diseases included individuals who suffered from arterial hypertension (43/300–14.33%), whereas people with insomnia constituted only 8.67% (26/300) of all respondents. The vast majority of respondents lived in the city (255/300—85.00%), and only 33.67% (101/300) did not engage in any physical activity. Detailed information on the demographic results obtained is presented in Table 1.

The results of the multivariate linear regression proved that, among the sociodemographic factors in the Karolinska Sleepiness Scale analysis, sleep disorders most often occurred in men (95%: 0.02; CI: 0.038; *p* < 0.001), in divorced persons (95%: 0.02; CI: 1.436; *p* = 0.045), and in overweight individuals (95%: 0.131; CI: 0.122; *p* < 0.001). The detailed data are presented in the Appendix A in Table A1.

A gender correlation with sleep disorders was also confirmed by the AIS (95%: 0.083; CI: 1.927; *p* = 0.026). These data are presented in the Appendix A in Table A2. The regression model used in the Epworth Scale analysis showed that the independent predictors of sleep disorders among nurses working in a shift system were living in the countryside (95%: −0.392; CI: −0.057; *p* = 0.009) and having children (95%: −0.027; CI: −0.058; *p* < 0.003). The results are presented in the Appendix A in Table A3.

### 3.1. The Work of Nurses and Sleepiness

The average work experience in the nursing profession was 15.95 ± 9.69 years and ranged from 0 to 38 years. A total of 120 of the 300 (40%) respondents had two jobs, and 18/300 (6%) of the respondents had three or more jobs. The analysis of the Karolinska Insomnia Scale data using the Kruskal–Wallis test showed that working in three or more workplaces was a factor influencing sleep disorders among the surveyed nurses (*p* = 0.005). The duration of night work averaged 13.73 ± 9.47 years and ranged from 0 to 38 years. Among the respondents, 47.33% (142/300) had the opportunity to rest after a night shift and 45.33% (136/300) only rested occasionally. Most of the respondents (183/300—61.00%) slept about 6–8 h on a work night, and only 24% (72/300) of the surveyed individuals said that they slept 5 h or less. As many as 81.33% (244/300) of the respondents were dissatisfied with their salary, while half of the respondents (155/300—51.67%) were satisfied with their work. Almost half of the respondents (142/300—47.33%) stated that they would continue to work. The results are presented in Table 2.

The results of the multivariate linear regression showed that, among the factors related to the work environment, the Epworth Scale analysis showed that night work (95%: 0.059; CI: 0.408; *p* = 0.009) was an independent predictor of sleep disorders among the nurses who worked shifts. Being employed at two workplaces was also an independent predictor (95%: 0.88; CI: 4.44; *p* = 0.004). The detailed information is presented in the Appendix A in Table A3.

### 3.2. The Quality of Sleep

Very severe sleep problems occurred in 1.67% (5/300) of the respondents, and moderate sleep problems occurred in 28.67% (86/300), while 29.67% (89/300) of the respondents said that they did not have any sleep problems. The most numerous group included individuals who were slightly worried about their health, and they constituted 41.67% (125/300) of all of the respondents. Among the respondents, 85% (255/300) did not use any drugs to facilitate falling asleep, and only four respondents used a sleep aid every day (1.33%). Of the surveyed individuals, 31.67% (95/300) watched TV or read a book before going to bed every day. Of the respondents, 35.67% (107/300) reported waking up when sleeping, while 21.00% (63/300) of the respondents reported continuous snoring when they slept. Additionally, states of confusion, anxiety, and agitation when waking up at night were reported by 38.33% (115) of the respondents. Furthermore, 46% (or 138/300) of the surveyed participants said that they did not feel rested after a night’s sleep. A mildly impaired psychophysical fitness was reported by 39.00% (119/300) of the respondents, while impaired psychophysical fitness was reported by 19.00% (57/300), clearly impaired psychophysical fitness was reported by 21.00% (63/300), and 6.67% (20/300) reported that they were not able to function normally. Half of the respondents (53.33%—160/300) fell asleep during the day on a day off work. The obtained results are presented in Table 3.

### 3.3. The Maslach Burnout Inventory and Sleepiness

The measurements using the MBI questionnaire showed that the average burnout score was 41.64 ± 24.25 points out of a possible 100 and ranged from 0 to 100 points. Emotional exhaustion (48.87 ± 31.84 points) was found to be most responsible for the occupational burnout of the respondents, while a lack of professional satisfaction was found to a slightly lesser extent (40.25 ± 27.01 points), and depersonalization was the least responsible (35.78 ± 32.57 points).

The analysis of the KSS questionnaire found that 16.67% of the respondents had problems with sleep (a score of ≥7 points). Seventy-three respondents (24.33%) had symptoms of somnolence when completing the questionnaire. However, the analysis of the Athens Insomnia Scale (AIS) questionnaire showed that as many as 73.33% (220/300) of the respondents suffered from insomnia. Additionally, the analysis of the Epworth Scale (ESS) questionnaire showed that 194/300 participants (64.67%) did not experience somnolence, 59/300 respondents (19.67%) had mild sleepiness, 31/300 respondents (10.33%) had moderate sleepiness, and 16 respondents (5.33%) had severe somnolence. According to the Spearman test, there was a correlation was between occupational burnout and the KSS, AIS, and ESS scores (*r* = 0.363, *r* = 0.518, and *r* = 0.345; *p* < 0.001). According to the Spearman correlation analysis, it was found that an increase in sleep disorders and occupational burnout was influenced by emotional exhaustion (KSS: *r* = 0.313, *p* < 0.001; AIS: *r* = 0.566, *p* < 0.001; ESS: *r* = 0.339, *p* < 0.001), depersonalization (KSS: *r* = 0.321, *p* < 0.001; AIS: *r* = 0.323, *p* < 0.001; ESS: *r* = 0.249, *p* < 0.001), and job satisfaction (KSS: *r* = 0.265, *p* < 0.001; AIS: *r* = 0.334, *p* < 0.001; ESS: *r* = 0.234, *p* < 0.001). The obtained results are presented in Table 4.

The multivariate linear regression result showed that, according to the KSS analysis, sleep disorders (95%: 1.368; CI: 5.564; *p* = 0.001), AIS (95%: 1.153; CI: 10.07; *p* = 0.014), and ESS (95%: 1.368); CI: 5.564; *p* = 0.001) were independent predictors of occupational burnout among the nurses who worked shifts. Details of the obtained data are presented in the Appendix A in Table A1, Table A2 and Table A3.

## 4. Discussion

The aim of our study was to determine the effect of demographic factors in nurses on their sleep disorders and occupational burnout. The study group included 300 nurses who work in a shift system. Among the sociodemographic factors, in the KSS analysis, sleep disorders were most common in men (CI: 0.038; *p* < 0.001), in divorced individuals (CI: 1.436; *p* = 0.045), and in individuals who were overweight (CI: 1.927; *p* = 0.026). Multiple linear regression showed that sleep disorders (*p* < 0.001) were an independent predictor of MBI among nurses who worked shifts.

Although the results of this empirical research are essentially consistent with what is presented in the literature on the subject, several separate, interesting observations were made. Zion et al. noted that the influence of bio-psychosocial factors on sleep disturbance among nurses is complex and depends on the interplay between these factors [33]. Research by these authors, who also used the Carolingian Sleepiness Scale, showed that sleep disturbances were related to the age of nurses, having children, and a chronotype of a preference for going to bed early and waking up early. The analyses in our research showed similar results, where, among the sociodemographic factors, sleep disorders most often occurred in people with children. On the other hand, the age of the surveyed nurses did not affect sleep disorders, and we did not study the behavior patterns of the nurses for the morning chronotype. According to Reinke et al., having children who are younger than 12 affects one’s tolerance for shift work, regardless of chronotype [34]. Additionally, the authors showed that nurses with the morning chronotype who took a short nap before the night shift adapted to shift work more easily.

In a study by Wisetborisut et al., it was shown that burnout occurs more frequently among shift workers compared to people who do not work at night [35]. In this study, shift workers who had 6–8 h of sleep a day and at least 8 days off a month had fewer burnout symptoms. This is in line with the analyses in our research where sleep disturbances were associated with night work and the professional burnout of nurses.

On the other hand, Jamal and Baba found no correlation between nurses working different shifts with occupational burnout in their studies, although they did observe that nurses working the night shift had a lower level of job satisfaction than their colleagues who did not work the night shift [36]. Similar results of a low level of job satisfaction were obtained in this study, where the lack of job satisfaction was associated with sleep disorders and the occupational burnout of the surveyed nurses.

A low level of job satisfaction might be related to the feeling of exhaustion from work. This was confirmed by the research of Młynarska et al. in which the MBI and MFIS (Modified Fatigue Impact Scale) occupational burnout questionnaire used was determined to be a useful tool for diagnosing these relationships. According to these authors, burnout among anesthesiology nurses who worked shifts was influenced by general fatigue (*p* = 0.017), as well as by physical (*p* = 0.017) and mental fatigue (*p* = 0.013), and sleep disturbances occurred in 37.33% of the respondents [37].

Fatigue immediately after work should be fully compensated for by rest and appropriate stress management [38]. Sleep is a natural fatigue reducer and fails to function when there is an imbalance between the sleep and wake mechanisms, which is quite common among the working population. Fatigue and difficulty sleeping are terms that are often used interchangeably [39]. The feeling of tiredness with one’s professional duties increases with the number of jobs one has. In this study, an analysis of data according to the Karolinska Insomnia Scale showed that working in three or more places was a significant factor that influenced sleep disorders among the studied nurses (*p* = 0.005). Working more than one job can be caused by a poor financial situation due to low unsatisfactory earnings and the desire to improve one’s financial situation. Taking up additional employment might also be favored by the nature of shift work because of the possibility for nurses to combine different jobs. In the presented study, as many as 81.33% of respondents were dissatisfied with their remuneration for with work, and sleep disorders mainly occurred in people who had additional jobs (*p* = 0.004). Moreover, in the study by Fontova-Almató et al., the vast majority of respondents were dissatisfied with their remuneration for their work, which had a direct significant impact on their lack of satisfaction with the work they performed (*p* = 0.034) [40]. According to Anabar et al., job satisfaction not only causes a low level of job satisfaction, but also contributes to emotional exhaustion [41]. This is in line with the results of our research in which the vast majority of respondents were dissatisfied with their salary, and emotional exhaustion, job dissatisfaction, and depersonalization were significant factors that influenced sleep disorders according to all three scales (KSS, AIS, and ESS).

In this study, it was observed that emotional exhaustion, which causes burnout, significantly affected the occurrence of sleep problems. Exhaustion can make work at night more tiring, while the body’s ability to regenerate worsens, which may directly cause health problems. In this study, 14.33% of the participants suffered from arterial hypertension, while diagnosed insomnia occurred in 8.67% of the participants. According to the results of a metanalysis that was conducted by Manohar et al., shift work plays an important role in hypertension; however, they did not confirm a significant influence of night work on sleep disorders [8]. Working at night disrupts all of the processes inside the body, as well as causes an increase in the stress hormones adrenaline and cortisol, which are the main representatives of glucocorticoid hormones. Numerous scientific studies have confirmed that sleep deprivation leads to an increase in the level of cortisol in the blood serum, which consequently puts the body in a state of constant readiness. As a result of this phenomenon, higher levels of this hormone cannot return to normal levels despite resting at night. Additionally, this process increases through the repetition of irregular rest phases. Interesting reports on this subject were presented by Cheungpasitporni et al., who observed that people who take naps during the day are more prone to developing arterial hypertension [42]. Additionally, in the study by Liu et al., it was observed that, among people who take naps during the day, there is a significant risk of a fatal cardiovascular incidents [43]. This was also confirmed by the research of Zhon et al., who also showed that people who take naps during the day have a significant risk of developing cancer [44]. On the other hand, the results of this study showed that as many as 53.33% of the nurses who work in a shift system take naps on their days off.

In the conducted studies, each additional kilogram of body weight according to BMI was an independent predictor of sleep disorders among the studied nurses who work in a shift system, despite the fact that nearly half of the respondents had a normal body weight. This is in line with the research of Sun et al., who showed that night shift work contributes to an increased risk of becoming overweight or obese, especially in the abdomen [45]. Other studies that were conducted among nurses suggest that night work, even permanent night work, mainly contributes to sleep disturbance [46].

According to the research presented here, the level of occupational burnout among the nurses who work in a shift system may be influenced by sleep disorders, and this relationship was the same in all of the surveys of sleep disorders that were used. These results are confirmed by the research of Giorgi et al., who showed that there is a correlation between occupational burnout and sleep quality among nurses who work shifts [47]. According to these authors, impaired sleep quality was more common among females, which differs from the results of the KSS analysis in our study where sleep disorders were more common in males. According to Vidotti et al., sleep disorders significantly affect the level of occupational burnout among nurses even among nurses who only work one shift [48]. On the basis of the above-cited research presented in the literature on the subject, as well as our own research, it can be said that it is not so much the shift work as the sleep disorders that significantly affect burnout among nurses.

### 4.1. Implications for Nursing Practice

The practical aim of this research was to indicate the directions and need for activities, the implementation of which could contribute to counteracting the phenomenon of burnout and sleep disorders among nursing staff. This knowledge will encourage the development of strategies to prevent burnout and sleep disorders, which will also improve the quality of nursing services. Recognizing the most important factors that determine occupational burnout among nursing staff can significantly contribute to counteracting this unfavorable phenomenon by taking targeted remedial actions. Therefore, there should be discussions, training on the impact of night work on the body, and an increase in remuneration for nurses who work the night shift. An effective system for monitoring shift schedules should be developed to reduce the level of sleep disturbances and burnout. Controlling the number of night shifts, prohibiting nurses from continuing to work during the day after the end of night duty, and observing appropriate long breaks for resting could all contribute to minimizing the negative effects of night work, which is the main cause of insomnia and sleep disorders in people with many years of shift work in the environment nursing.

### 4.2. Limitations of the Study

The limitations of this work include the small size of the study group (*n* = 300) and the selection of nurses from only one region in Poland (Silesian Region). Therefore, care should be taken in assigning these results to the profile of all nurses nationwide and worldwide. Because this research needs to be extended in the future in order to expand the knowledge in this area, we plan to include more study groups in a multicenter study. Replicating this study with more data and more robust study designs is warranted in order to confirm these results.

## 5. Conclusions

The sociodemographic factors that influence sleep disorders in nurses who work shifts include male gender, divorce, living in the countryside, and having children. Among the factors related to the work environment that influence sleep disorders are night work and having professional activity in two workplaces. Sleep disorders cause the burnout of nurses who work shifts. This knowledge can encourage the development of strategies to prevent burnout and sleep disorders, which will also improve the quality of nursing services.

## Figures and Tables

**Table 1 ijerph-19-06218-t001:** The correlations between sleepiness scales and characteristics of the study participants.

Variable		N = 300	KSS	AIS	ESS
Gender *n* (%), *M* ± *SD*	*p*-value ^1^		0.054	0.313	0.089
	Females	264 (88.00%)	4.73 ± 1.86	8.54 ± 4.38	9.13 ± 5.29
	Males	35 (11.67%)	4.17 ± 2.44	7.89 ± 5.22	7.57 ± 5.02
Age (years) *n* (%)	*p*-value ^2^		0.682	0.187	0.439
	20–30	56 (18.67%)			
	31–40	98 (32.67%)			
	41–50	117 (39.00%)			
	51–60	25 (8.33%)	*r* = 0.024	*r* = 0.077	*r* = −0.045
	61–70	1 (0.33%)			
	No answer	3 (1.00%)			
BMI (kg/m^2^) *n* (%)	*p*-value ^2^		0.242	0.741	0.576
	Emaciation	2(0.67%)			
	Underweight	5 (1.67%)			
	Normal	158 (52.67%)			
	Overweight	90 (30.00%)	*r* = 0.068	*r* = 0.019	*r* = 0.033
	Obesity	30 (10.00%)			
	Obesity II°	12 (4.00%)			
Marital status *n* (%), *M* ± *SD*	*p*-value ^3^		0.068	0.018 *	0.004 *
	Single,	67 (22.33%)	4.48 ± 1.97	7.6 ± 4.01	8.46 ± 4.57
	Married	203 (67.67%)	4.6 ± 1.85	8.4 ± 4.33	8.62 ± 5.24
	Divorced	26 (8.67%)	5.53 ± 2.24	10.87 ± 5.6	12.23 ± 5.88
	Widowed	4 (1.33%)			
Education *n* (%), *M* ± *SD*	*p*-value ^3^		0.283	0.398	0.25
	Medium	26 (8.67%)	4.19 ± 1.96	8.62 ± 4.36	7.42 ± 4.97
	Bachelor’s degree	135 (44.33%)	4.59 ± 1.89	8.04 ± 4.26	8.82 ± 4.9
	Master’s degree	133 (44.33%)	4.8 ± 1.97	8.78 ± 4.65	9.27 ± 5.6
	Doctorate	3 (1.00%)			
	Other,	2 (0.67%)			
	No answer	1 (0.33%)			
Children *n* (%), *M* ± *SD*	*p*-value ^1^		0.275	0.68	0.179
	Yes, *n* (%)	209 (69.67%)	4.59 ± 1.89	8.5 ± 4.22	9.26 ± 5.48
	No, *n* (%)	91 (30.33%)	4.84 ± 2.05	8.4 ± 5.03	8.22 ± 4.7
	Preschool-age children	154 (73.84%)			
	Children attending nursery/kindergarten	52 (24.88%)			
Physical activity *n* (%)	Daily	17 (6.67%)			
	A few times a week	71 (23.67%)			
	A few times a month	110 (36.67%)			
	No	101 (33.67%)			
	No answer	1 (0.33%)			
Place of residence *n* (%)	*p*-value ^1^		0.987	0.789	0.56
	Urban	255 (85.00%)	4.67 ± 1.92	8.44 ± 4.41	9.04 ± 5.29
	Rural	43 (14.33%)	4.63 ± 2.03	8.65 ± 4.96	8.35 ± 517
	No answer	2 (0.67%)			

^1^ Mann–Whitney test. ^2^ Spearman’s correlation coefficient. ^3^ Kruskal–Wallis test with post hoc analysis (Dunn test). BMI—body mass index; KSS—Karolinska Sleepiness Scale; AIS—Athens Insomnia Scale; ESS—Epworth Sleepiness Scale. * statistically significant relationship (*p* < 0.05).

**Table 2 ijerph-19-06218-t002:** Characteristics of work and sleepiness of the nurses.

Variables	N = 300	KSS	AIS	ESS
Seniority as a nurse (years), *p*-value ^1^		0.74	0.208	0.594
0–10, *n* (%)	108 (36.00%)			
11–20, *n* (%)	94 (31.33%)	*r* = 0.019	*r* = 0.073	*r* = −0.031
21–30, *n* (%)	79 (26.33%)			
31–40, *n* (%)	19 (6.33%)			
Work in the night shift (years), *p*-value ^2^		0.932	0.237	0.188
0–10, *n* (%)	132 (44.00%)	*r* = 0.005	*r* = 0.069	*r* = −0.077
11–20, *n* (%)	98 (32.67%)			
21–30, *n* (%)	55 (18.33%)			
31–40, *n* (%)	12 (4.00%)			
No answer, *n* (%)	3 (1.00%)			
Number of jobs held, *p*-value ^2^		0.005 *	0.314	0.108
One, *n* (%), *M* ± *SD*	158 (52.67%)	4.44 ± 1.94	8.14 ± 4.41	8.47 ± 5.02
Two, *n* (%), *M* ± *SD*	120 (40.00%)	4.72 ± 1.84	8.48 ± 3.98	9.08 ± 5.25
>Three, *n* (%), *M* ± *SD*	18 (6.00%)	6.11 ± 2.05	10.34 ± 6.83	11.56 ± 6.23
No answer, *n* (%), *M* ± *SD*	4 (1.33%)			
Work satisfaction				
Yes, *n* (%)	155 (51.67%)			
Medium, *n* (%)	129 (43%)			
No, *n* (%)	16 (5.33%)			
Change of shift work to one shift				
Yes, *n* (%)	142 (47.33%)			
No, *n* (%)	155 (47.33%)			
No answer, *n* (%)	3 (1.00%)			
Rest after night shift				
No, *n* (%)	16 (5.33%)			
Rarely, *n* (%)	136 (45.33%)			
Yes, *n* (%)	142 (47.33%)			
No answer, *n* (%)	6 (2.99%)			
Length of sleep without being on night shift				
<5 h, *n* (%)	72(24.00%)			
6–8 h, *n* (%)	183 (61.00%)			
>8 h, *n* (%)	41 (13.67%)			
No answer, *n* (%)	4 (1.33%)			

^1^ Spearman’s correlation coefficient. ^2^ Kruskal–Wallis test with post hoc analysis (Dunn test). * statistically significant relationship (*p* < 0.05). KSS—Karolinska Sleepiness Scale; AIS—Athens Insomnia Scale; ESS—Epworth Sleepiness Scale.

**Table 3 ijerph-19-06218-t003:** Characteristics of the quality of sleep.

Variable	N = 300	Data
Occurrence of sleep problems, *n* (%)	No	89 (29.67%)
	Mild	96 (32.00%)
	Moderate	86 (28.67%)
	Heavy	24 (8.00%)
	Very heavy	5 (1.67%)
Worry about health, *n* (%)	No	56 (18.67%)
	Slightly	125 (41.67%)
	In a measure	92 (30.67%)
	Significantly	18 (6.00%)
	Very excited	8 (2.67%)
	No answer	1 (0.33%)
Taking medicines to help you fall asleep, *n* (%)	Daily	4 (1.33%)
	A few times a week	10 (3.33%)
	A few times a month	31 (10.33%)
	No	255 (85.00%)
Watching TV and/or reading books before going to bed, *n* (%)	Daily,	95 (31.67%)
	Every other day	19 (6.33%)
	Rarely	114 (38.00%)
	No	62 (20.67%)
Waking up at night, *n* (%)	No	37 (12.33%)
	Rarely	156 (52.00%)
	Often	107 (35.67%)
State of confusion and anxiety during waking up, *n* (%)	A few times a week	13 (4.33%)
	A few times a month	102 (34.00%)
	No	159 (53.00%)
	Always after a sleepless night before	26 (8.67%)
Snoring while sleeping, *n* (%)	No	148 (49.33%)
	Rarely	89 (29.67%)
	Yes	63 (21.00%)
Feeling rested after waking up, *n* (%)	Yes	48 (16.00%)
	Still tired	65 (21.67%)
	No energy to act	38 (12.67%)
	Lack of strength to get out of bed	11 (3.67%)
	I would sleep longer, *n* (%)	138 (46.00%)
Naps on nonworking days, *n* (%)	Yes	160 (53.33%)
	No	140 (46.67%)

**Table 4 ijerph-19-06218-t004:** The correlations between sleepiness and burnout.

Variable	N = 300	MBI	1	2	3
Karolinska Sleepiness Scale, *n* (%)					
Extraordinary wakefulness	11 (3.67%)				
Extra wakefulness	26 (8.67%)				
Wakefulness	66 (22.00%)				
Rather wakefulness	48 (16.00%)				
Neither wakefulness nor sleepy	26 (8.67%)				
I have signs of sleepiness	73 (24.33%)				
Sleepy but I easily resist sleepiness	28 (9.33%)				
Sleepy but I have difficulty resisting sleepiness	15 (5.00%)	*r* = 0.363	*r* = 0.313	*r* = 0.321	*r* = 0.265
Extremely sleepy	7 (2.33%)	*p <* 0.001 *	*p <* 0.001 *	*p <* 0.001 *	*p <* 0.001 *
Athenian Insomnia Scale, *n* (%)					
No insomnia	80 (26.67%)	*r* = 0.518	*r* = 0.566	*r* = 0.323	*r* = 0.334
Insomnia	220 (73.33%)	*p <* 0.001 *	*p <* 0.001 *	*p <* 0.001 *	*p <* 0.001 *
The Epworth Scale Sleepiness, *n* (%)					
Result is correct	194 (64.67%)				
Mild sleepiness	59 (19.67%)	*r* = 0.345	*r* = 0.339	*r* = 0.249	*r* = 0.234
Moderate sleepiness	31 (10.33%)	*p <* 0.001 *	*p <* 0.001 *	*p <* 0.001 *	*p <* 0.001 *
Heavy sleepiness	16 (5.33%)				

MBI—Maslach Burnout Inventory. *p*-Value—Spearman’s correlation coefficient. * Statistically significant relationship (*p* < 0.05). 1—subscale MBI: exhaustion; 2—subscale MBI: depersonalization; 3—subscale MBI: job dissatisfaction.

## Data Availability

The data presented in the study are available on request from the first and second authors. The data are not publicly available due to ethical and privacy restrictions.

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
