# Peer review of "Determinants of Sleep Disorders and Occupational Burnout among Nurses: A Cross-Sectional Study"

_ijerph, 2022, doi:10.3390/ijerph19106218_

Round 1

Reviewer 1 Report

Shift work could cause sleep disturbance. This study explored the determinants of sleep disorders and burnout among nurses. I have several comments for consideration.

Introduction:

The authors highlighted the impact of sleep disorders. However, it was unclear the relationship between sleep disorders and burnout. Please add some content related to burnout in this section.

Materials and methods:

  1. Nurses on shift were included in this study. Please make operational definition of "shift". For example, nurse worked after 6pm was on shift.
  2. How would you determine the sample size? How would you recruit the participants?
  3. Regarding the questionnaires, please show their psychometric property as well as the number of items and rating method.
  4. Was there an open-ended questions? I was not clear what 'qualitative variables' would be obtained from the questionnaires.

Discussion

  1. Please add a summary of the findings at the beginning.
  2. Implications were broad. Please give some practical suggestions with evidence support. For example, what strategies from available studies were shown to prevent burnout and sleep disorders?

Author Response

We appreciate a detailed revision that finalny improved the quality of this paper and the great effort chich Has been done by Reviewer 1. We wolud like to ensure that we have adressed all comments In our point-by point reply as below.

Comments and Suggestions for Authors

Shift work could cause sleep disturbance. This study explored the determinants of sleep disorders and burnout among nurses. I have several comments for consideration.

Introduction:

The authors highlighted the impact of sleep disorders. However, it was unclear the relationship between sleep disorders and burnout. Please add some content related to burnout in this section.

- Reply:

Thank you for your comments  which will improve the incroduction as it was suggested.

Materials and methods:

  1. Nurses on shift were included in this study. Please make operational definition of "shift". For example, nurse worked after 6pm was on shift.

-Reply:

Thank you for you suggestion, we added the characteristic of shif work in the text:

In nursing, there are many varieties of shift work organization systems, such as three-shift, two-shift, ten-hour etc. In many clinics, nurses work in one shift, that is, from Monday to Friday in equal hours. In turn, section nurses working in hospitals usually perform their work in a two-shift system. It is characterized by two 12-hour shifts (day and night). Nurses start work at 7 a.m. and  7 p.m. On average, they work in one shift for one day, with two days off followed by two days off.”

  1. How would you determine the sample size? How would you recruit the participants?

-Reply:

Thank you for yours comments we improved our manuscript and we put the sample i the text:

„The minimum sample size was 266, which was calculated based on the available nurses population with a 95% confidence interval. The data to determine the minimum number of individuals was obtained from the  based on a report by the Supreme Chamber of Nurses in Poland”.

  1. Regarding the questionnaires, please show their psychometric property as well as the number of items and rating method.

-Reply:

Thank you for your suggestion we improved the methods section.

  1. Was there an open-ended questions? I was not clear what 'qualitative variables' would be obtained from the questionnaires.

-Reply:

Thank you for yours comments. The questionnaires were distributed by nursing leaders to the medical staff in the department. Surveys in which not all responses were recorded were rejected.

Discussion

  1. Please add a summary of the findings at the beginning.

-Reply:

Thank you for your comments  which will improve our work. As suggested by the review, we added a summary of the findings at the beginning the discution section.

  1. Implications were broad. Please give some practical suggestions with evidence support. For example, what strategies from available studies were shown to prevent burnout and sleep disorders?

-Reply:

Thank you for the suggestion, we corrected a certain inaccuracy in the sentence:

"This knowledge will enable strategies to prevent burnout and sleep disorders to be developed in order to maintain and improve the quality of nursing services".

on

"This knowledge will encourage the development of strategies to prevent burnout and sleep disorders, which will also improve the quality of nursing services".

Please, kindly  noted, that in lines 338 -346 the proposed specific interventions (strategies) are listed. A review of the literature on strategies to prevent burnout and sleep disorders was not the aim of this study.

Reviewer 2 Report

  1. This is a well-written manuscript.
  2. The introduction is relevant and theory-based.
  3. Sufficient information about the previous study.
  4. The methods are generally appropriate.
  5. The results are clear.
  6. The authors make a systematic contribution to the research literature in this area of investigation.
  7. Overall, this is a quality manuscript that has implications for the theoretical basis and development of determinants of sleep disorders and occupational burnout among nurses.
  8. This work is also a good reminder for administrations to pay attention to the effects of sleep disorders and occupational burnout among nurses.
  9. The conclusion could be strengthened by adding some objective statements about the contributions of this study as a reference for the field.

Author Response

We appreciate a detailed revision, notes comments and the great effort which has been done by Reviewer 2.

  1. This is a well-written manuscript.
  2. The introduction is relevant and theory-based.
  3. Sufficient information about the previous study.
  4. The methods are generally appropriate.
  5. The results are clear.
  6. The authors make a systematic contribution to the research literature in this area of investigation.
  7. Overall, this is a quality manuscript that has implications for the theoretical basis and development of determinants of sleep disorders and occupational burnout among nurses.
  8. This work is also a good reminder for administrations to pay attention to the effects of sleep disorders and occupational burnout among nurses.
  9. The conclusion could be strengthened by adding some objective statements about the contributions of this study as a reference for the field.

-Reply:

Thank you for yours suggestion. We improved our manuscript.

Round 2

Reviewer 1 Report

The authors addressed all my comments well. I have no new comment raised after reading.